# Correlation of HPV Status with Colposcopy and Cervical Biopsy Results Among Non-Vaccinated Women: Findings from a Tertiary Care Hospital in Kazakhstan

**DOI:** 10.3390/vaccines13111151

**Published:** 2025-11-11

**Authors:** Talshyn Ukybassova, Gulzhanat Aimagambetova, Kuralay Kongrtay, Kuat Kassymbek, Milan Terzic, Sanimkul Makhambetova, Makhabbat Galym, Nazira Kamzayeva

**Affiliations:** 1Clinical Academic Department of Women’s Health, CF “University Medical Center”, Astana 010000, Kazakhstan; talshynu@yandex.ru (T.U.); kkongrtay@nu.edu.kz (K.K.); sanai.nadirova@gmail.com (S.M.); mahabbat_g@mail.ru (M.G.); nazira.kamzaeva@umc.org.kz (N.K.); 2Department of Surgery, School of Medicine, Nazarbayev University, Astana 010000, Kazakhstan

**Keywords:** human papillomavirus, cervical intraepithelial neoplasm, squamous intraepithelial lesions, Kazakhstan

## Abstract

**Background/Objectives**: Cervical cancer is one of the most frequent malignancies among women in Kazakhstan, where human papillomavirus (HPV) vaccination was initiated in 2024. Despite the implementation of vaccination and cytology-based screening programs, diagnostic limitations remain, and local evidence linking HPV infection to clinical outcomes is scarce. This study aimed to evaluate the correlation between HPV status, cervical cytology results, colposcopic impression, and biopsy results in a non-vaccinated female population. **Methods**: A cross-sectional study was conducted at the University Medical Center, Astana, between November 2024 and March 2025. A total of 396 women of reproductive age were enrolled. Cervical samples underwent liquid-based cytology and high-risk HPV testing with the RealBest assay. Colposcopy was performed following abnormal cervical cytology results, and colposcopy-guided biopsies were obtained where indicated. Sociodemographic characteristics were assessed, and associations between HPV genotype and clinical outcomes were analyzed using descriptive and inferential statistics. **Results**: HPV infection was detected in 140 women (35.4%). HPV-16 was the most common genotype (11.4%), followed by HPV-52 (6.6%) and HPV-33 (5.3%). Among 198 women evaluated by colposcopy, abnormal findings were observed in 72.2%, with HPV-16 showing a significant association with higher-grade abnormalities (*p* < 0.001). Biopsies were available for 40 participants: 12 had CIN I, 12 had CIN II, 10 had CIN III, and 4 had carcinoma *in situ*. HPV-16 was the only genotype significantly linked to CIN II/III lesions. **Conclusions**: HPV-16 was strongly associated with abnormal colposcopic findings and high-grade histology, underscoring its oncogenic importance. The prevalence of HPV-52 and HPV-33 further supports the need for HPV nonavalent vaccination. These findings highlight the importance of HPV-based screening, genotype-specific triage, and expanded vaccination to reduce cervical cancer incidence in Kazakhstan.

## 1. Introduction

Cervical cancer continues to be one of the most common female malignancies among women worldwide and remains a leading cause of cancer-related mortality in many low- and middle-income countries [1]. Although cervical cancer is one of the preventable cancers through the appropriate implementation of human papillomavirus (HPV) vaccination and screening, the incidence of cervical cancer is increasing globally. It is estimated that 703,000 new cervical cancer cases will be confirmed by the end of 2025, and the mortality will reach 373,000 [2,3,4]. In Kazakhstan, cervical cancer ranks among the top three cancers affecting women, with an estimated age-standardized incidence rate of approximately 19 per 100,000 with increasing trends over the past decade, while mortality is stable (around 600 cases per year) [5,6,7]. A nationwide cytology-based cervical cancer screening program was introduced in Kazakhstan in 2017 and fully covered by the state budget. The target group is women aged 30–70 years, with the screening to be performed every four years [8]. The screening relies on liquid-based cytology reported according to the Bethesda system, and abnormal findings are followed by colposcopic examination and biopsy, where indicated. While the cervical cancer screening program is well established, participation rates vary, and challenges persist in terms of coverage, diagnostic accuracy, and follow-up care [9,10]. The screening coverage remains low (46%) and does not reach the target level proposed by the World Health Organization (WHO) [11]. Recent analyses have emphasized the need to strengthen program quality, particularly considering global trends toward HPV-based primary screening [12].

Primary prevention strategies have also been evolving. A pilot HPV vaccination initiative was attempted in 2013 but was discontinued due to public concerns and limited acceptance [13]. In 2024, Kazakhstan reintroduced HPV vaccination at the national level, targeting girls aged 11–13 years through school-based delivery, with the aim of increasing coverage in line with the WHO Cervical Cancer Elimination Initiative [14]. The HPV vaccination program in Kazakhstan is covered by the state budget and is free of charge for patients. Monitoring HPV genotype distribution is particularly important in this transitional period, as high-risk types HPV-16, -18, -31, -33, and -35 remain the most prevalent among women in Kazakhstan [15,16]. Even though the national HPV vaccination program was re-launched, the effect of vaccination in terms of decreasing the incidence of HPV infection and cervical cancer will be detectable in the coming 10 years [17].

Colposcopy and biopsy play central roles within the screening pathway, providing diagnostic confirmation after abnormal screening results and guiding treatment decisions. However, the strength of association between HPV genotype, cytological abnormalities, colposcopic impression, and histopathological outcomes may differ by setting, reflecting variations in population epidemiology and clinical practice [18,19]. Local data are therefore critical for optimizing diagnostic algorithms. Moreover, previous studies in Kazakhstan have described the distribution of HPV genotypes and examined screening practices and attitudes toward vaccination [20,21]; however, there are no studies investigating the correlation of HPV status with cervical cytology or colposcopy results. Therefore, the aim of this study was to investigate the correlation between HPV status, cervical cytology, colposcopic findings, and biopsy results among women in Kazakhstan. The results of the study are expected to support improvements in cervical cancer screening, HPV vaccination, and management of cervical cancer in Kazakhstan.

## 2. Materials and Methods

### 2.1. Study Design and Study Subjects

A cross-sectional study was conducted in the University Medical Center (UMC). UMC is a tertiary care hospital in Astana (Kazakhstan), which is a part of the Nazarbayev University Medical Cluster, consisting of multiple clinical settings, including the National Research Center for Mother and Child Health (Mother and Child Hospital). The Mother and Child Hospital is caring for all patients who voluntarily aim to be examined in its outpatient unit without age and condition restrictions, as there is a range of trained obstetrics and gynecology subspecialists, including adolescent gynecologists, oncogynecolosist, and gynecologic endocrinologists. Moreover, it accumulates patients from different areas of the country referred for examination to the tertiary care level clinic by the regional/rural physicians. The UMC in general, and the Mother and Child Hospital in particular, has a well-equipped outpatient unit that became a primary setting for this study (patient recruitment and examination).

During the study period (between November 2024 and March 2025), all women attending the outpatient gynecology clinic were first screened for eligibility by the attending clinician during routine consultation. Only those meeting all the inclusion criteria were then invited for the study. Of the 753 eligible women who were invited, 396 agreed to participate, with a participation rate of 52.6%.

Women who agreed to take part in the study underwent cervical smear collection and HPV testing for cervical screening and were referred to colposcopic examination and biopsy, if indicated (Figure 1). Overall, 753 women were screened in UMC and invited to take part in the study; however, only 396 agreed and were included in this research dataset.

The sample size calculation for estimating the proportion in a cross-sectional study was performed using the standard formula, which assumes a confidence level of 95% (Z = 1.96), estimated prevalence of HPV-associated precancerous cervical lesions to be 21% (*p* = 0.21), q = 1 − *p* = 0.79, and maximum sampling error of 5% (Δ = 0.05). For these parameters, the calculated minimum required sample size was 400 women. The prevalence of precancerous cervical lesions among women of reproductive age in Kazakhstan [11,21,22] was considered. The used equation is n = Z^2^pq/∆^2^, where

n—sample size,Z—coefficient depending on the confidence level chosen by the researcher,p—proportion of respondents with the presence of the studied characteristic,q = 1 − p—the proportion of respondents who do not have the studied characteristic,∆—maximum sampling error.

Women were included in the study if they met the following inclusion criteria: (1) age between 18 and 46 years old; (2) were not pregnant at the time of the study; (3) had a regular menstrual cycle; and (4) had an intact uterine cervix (did not have prior surgeries on the uterine cervix). Exclusion criteria were the following: (1) age younger than 18 and older than 46 years; (2) the presence of complex concomitant chronic diseases (hepatitis B and C, diabetes mellitus, autoimmune diseases, HIV-infected and oncological diseases at present and in history) in any location; (3) acute inflammatory processes of any localization at the time of the study; use of probiotics and/or antibiotic therapy and/or immunosuppressive therapy within the previous 14 days; (4) smoking; (5) intrauterine device *in situ*; (6) history of HPV vaccination; (7) any invasive procedures and surgical interventions on organs and genitals within 45 days preceding the study.

The recruitment took place in the UMC, Mother and Child Hospital, outpatient unit. All women (who met the inclusion and exclusion criteria) attending the clinic during the study period were invited to take part in the research. Clinical data of those patients who agreed to participate were collected after their informed consent.

The study participants’ clinical data were collected using a 22-item patient survey that enables the collection of the following information used as variables for the study statistical analyses: age, education, marital status, weight, height, body mass index (BMI), menarche, last menstrual cycle dates, contraception, number of sexual partners, history of gynecologic conditions and surgeries, prior cervical cancer screening tests/results, HPV status (if known), etc.

### 2.2. Human Papillomavirus Testing and Liquid-Based Cytology

Cervical specimens were obtained using an endocervical brush, sampling both the ectocervix and the endocervical canal, including the transformation zone where squamous and columnar epithelium meet. The collected samples were immediately placed into transport medium and delivered to the laboratory on the same day.

For HPV genotyping, cervical swabs were processed with the RealBest assay, which detects 12 oncogenic HPV types (16, 18, 31, 33, 35, 39, 45, 51, 52, 56, 58, and 59), in accordance with the manufacturer’s protocol. Real-time PCR amplification was performed on the CFX 96 RealTime PCR system (Bio-Rad Laboratories Inc., Hercules, CA, USA). Each run included positive and negative controls. DNA input was standardized at 3.75 ng/μL, corresponding to 37.5 ng per reaction well. Data were analyzed using the manufacturer’s software, and HPV type positivity was determined according to predefined thresholds.

Liquid-based cytology (LBC) Pap testing was carried out in the UMC laboratory among all study women. Cervical samples were collected by hospital gynecologists using standard techniques. A cytobrush was inserted into the external cervical os, rotated to collect cervical canal epithelial cells, and then placed into an LBC preservative fluid vial. This reagent eliminates excess waste material, such as mucus and blood, before exiting a stable coating of cells on slides. The vials were transported to the cytology laboratory, where samples were vortexed at 3000 rpm for 15–20 s to remove more mucosal and blood contaminants. After the addition of a density reagent, samples were sedimented and centrifuged at 2500× *g* for 5 min. Two washes with alcohol preceded Papanicolaou staining using routine methods.

Slides were read and examined under Axioscope 40 and Axiostar Plus microscopes (Zeiss, Jena, Germany). Cytological findings were reported and interpreted according to the Bethesda system for cervical cytology [22].

### 2.3. Colposcopy and Colposcopy-Guided Biopsy

The following equipment/consumables were used for the procedure—digital videocolposcope, vaginal speculum, 3% acetic acid, 5% iodine solution, and biopsy forceps. The colposcopy was performed using the MK-200 digital videocolposcope (Scaner, MK-200, SCANER Company, Cherkassy, Ukraine) with interchangeable magnifications from 3.3× to 22×.

A patient was appropriately positioned on a gynecological chair, and the uterine cervix was visualized per vaginal speculum. Visual inspection of the vagina and cervix was performed with the naked eye before colposcopic examination. Then videocolposcopy was performed with 3% acetic acid, which was applied to the vaginal part of the uterine cervix with a cotton swab and allowed to soak for 1 min; visual examination was performed with a colposcope. As the next step, 5% iodine solution was applied to the vaginal part of the uterine cervix to highlight the dysplastic areas (if any) [23].

Histopathological outcomes were graded according to the WHO terminology: normal, cervical intraepithelial neoplasia grade 1 (CIN1), cervical intraepithelial neoplasia grade 2 (CIN2), cervical intraepithelial neoplasia grade 3 (CIN3), and carcinoma *in situ* [24].

### 2.4. Statistical Analysis

Continuous variables were summarized as means with standard deviations or as medians with interquartile ranges, depending on their distribution. Group comparisons between HPV-positive and HPV-negative women were conducted using the independent samples *t*-test or the Mann–Whitney U test when distributional assumptions were not met. Categorical variables were presented as counts and percentages, and associations were assessed using the chi-square test or Fisher’s exact test, as appropriate. For ordinal outcomes, the Cochran–Armitage trend test was applied.

Pairwise post hoc comparisons were evaluated using odds ratios with 95% confidence intervals, calculated directly from 2 × 2 contingency tables, with Fisher’s exact test applied for significance testing and Holm correction for multiple comparisons. Effect sizes were reported using Cramér’s V for categorical associations. To account for multiple hypothesis testing across overall analyses, *p*-values were further adjusted using the false discovery rate (FDR) method (Benjamini–Hochberg procedure).

All statistical analyses were performed in Python version 3.12.11, using the libraries NumPy (2.0.2), Pandas (2.2.2), SciPy (1.16.2), rpy2 (3.5.17), Statsmodels (0.14.5), Scikit-learn (1.6.1), Matplotlib (3.10.0), and Seaborn (0.13.2).

### 2.5. Ethical Consideration

The study was conducted in accordance with the Declaration of Helsinki and its subsequent revisions. Written informed consent was obtained from all participants prior to sample collection. Ethical approval was granted by the Local Bioethics Committee of the “UMC” Corporate Fund (Minutes No. 2024/02-013, dated 10 May 2024). Throughout the study, the investigators adhered to established principles of biomedical research ethics and scientific integrity. No personally identifiable information was accessible to the research team at any stage of the study.

## 3. Results

### 3.1. Socio-Demographic Description of Study Subjects

A total of 396 non-vaccinated women were enrolled. The mean age of the cohort was 34.5 ± 6.4 years (range: 19–46). Sociodemographic details, including education level, marital status, and reproductive history, are presented in Table 1. The median age of women without HPV infection was 34.96 years (IQR: 31.60–40.00), which was significantly higher than that observed among HPV-positive women (median 33.86 (IQR: 27.92–37.48); *p* = 0.0019). Marital status also varied significantly between the two groups (*p* = 0.0078), with single women more frequently represented among HPV-positive participants (31.2%) compared with HPV-negative participants (18.8%). No significant differences were identified in other sociodemographic characteristics, including educational attainment, BMI, or mode of delivery (Table 1). With respect to reproductive and clinical history, HPV-positive women reported a lower number of pregnancies (mean 1.51 vs. 2.38; *p* < 0.001) and deliveries (mean 1.20 vs. 1.74; *p* = 0.0002) than HPV-negative women. Abortion history also differed significantly between groups (*p* = 0.0011). In addition, barrier contraceptive use was more commonly reported by HPV-positive women (41.0%) than by those without HPV infection (29.2%; *p* = 0.0232) (Table 1).

### 3.2. Human Papillomavirus Status of Study Subjects

Out of 396 participants, 256 women were negative for any HPV infection. Among high-risk HPV types detected, HPV-16 was the most prevalent, identified in 45 (11.4%) of women. Other frequently observed genotypes included HPV52, found in 26 women (6.6%), and HPV-33—found in 21 women (5.3%). The most frequent HPV coinfections were HPV56 and HPV33, simultaneously detected in 4 cases, as well as HPV18 and HPV52, found in 3 cases of coinfection. Overall, the distribution highlights the predominance of HPV16, followed by HPV-52 and HPV-33, underscoring their major role in the burden of high-risk HPV infections in this cohort (Figure 2).

Colposcopic evaluation was available for 198 women. Normal colposcopic appearance was documented in 48.5%, abnormal grade I in 40.1%, and abnormal grade II or higher in 10.6%. Analysis of HPV type distribution across colposcopic findings revealed a significant association between HPV-16 and abnormal colposcopic impression (Table 2). The prevalence of HPV-16 increased from 9.4% in women with normal colposcopy to 19.8% with grade I abnormalities and 52.4% with grade II abnormalities (χ^2^ = 21.65, χ^2^ FDR-adj. *p* < 0.001, CATT FDR-adj. *p* < 0.001). None of the other high-risk HPV types, including HPV-18, HPV-31, HPV-33, HPV-35, HPV-39, HPV-45, HPV-51, HPV-52, HPV-56, HPV-58, or HPV-59, showed statistically significant trends after correction for multiple testing. Overall HPV status (positive vs. negative) was not significantly associated with colposcopic grade (χ^2^ = 3.66, FDR-adj. *p* = 0.694), indicating that the effect was largely driven by HPV-16 (Table 2).

Biopsy results were available for 40 women. Histological assessment revealed cervicitis in 2 cases, CIN I in 12, CIN II in 12, CIN III in 10, and carcinoma *in situ* in 4 patients. When stratified as CINII+ versus <CINII, 26 women (65%) were classified as CINII+. HPV-16 was the most prevalent genotype, detected in 36.8% of cases, with frequency rising from 16.7% in CIN I to 80% in CIN III and 75% in cancer *in situ* (CIS), showing a moderate-to-strong effect before adjustment. Other high-risk types (HPV-18, HPV-33, HPV-35, HPV-39, HPV-51, HPV-52, HPV-56, and HPV-59) were infrequent (<8%) and not significantly associated with lesion grade. Notably, HPV-45 and HPV-58, though rare, were disproportionately present in CIS (50% each), with strong effect sizes but significance lost after false discovery rate adjustment. These findings confirm HPV-16 as the predominant driver of progressive cervical neoplasia, while HPV-45 and HPV-58 may contribute disproportionately to high-grade precancerous lesions in this cohort (Table 3).

The distribution of biopsy outcomes varied substantially by colposcopic grade. In the 2-grade impression colposcopy category, biopsy outcomes were predominantly high-grade lesions, with 12 cases of CIN II (57.1%), 4 cases of carcinoma *in situ* (19.0%), 3 cases of CIN III (14.3%), and 2 cases of CIN I (9.5%). No cases of cervicitis were observed, indicating a strong alignment between high-grade colposcopic findings and severe histological diagnoses. The 1-grade impression colposcopy category showed a more mixed distribution, with 10 cases of CIN II (45.5%), 9 cases of CIN I (40.9%), 2 cases of CIN III (9.1%), 1 case of cervicitis (4.5%), and no carcinoma *in situ*. This suggests that low-grade colposcopic impressions often correspond to moderate lesions, though a substantial proportion escalates to CIN II or higher upon biopsy. For the normal colposcopy category, the limited sample revealed 2 cases of CIN II (50.0%), 1 case of cervicitis (25.0%), 1 case of CIN I (25.0%), and no CIN III or carcinoma *in situ*. The presence of CIN II in half the cases highlights potential under-detection of moderate lesions in apparently normal colposcopies. Overall, the distributions demonstrate a clear trend of increasing lesion severity with higher colposcopy grades, though overlaps (e.g., CIN II in normal and low-grade categories) underscore the imperfect predictive value of colposcopy alone (Figure 3).

## 4. Discussion

Despite the fact that cervical cancer can be prevented by screening and HPV vaccination, the disease remains an important global health problem [1,3,4]. In Kazakhstan, cervical cancer is the second most common cancer among females, with an increasing incidence in the past 10 years [6,7]. Moreover, local studies identified a high prevalence of high-risk HPV types among women [15,16]. This investigation evaluated associations between HPV genotypes, cytology, colposcopy, and biopsy results in Kazakhstan in order to provide evidence to improve national cervical cancer prevention, screening, and management protocols.

This cross-sectional study demonstrated a strong association between high-risk HPV infection, particularly HPV16, colposcopic impression, and histologic outcomes among the non-vaccinated female population in Kazakhstan. Our findings align with international studies but provide unique local evidence where HPV vaccination coverage was extremely low (<2%) before 2024, when the nationwide HPV vaccination campaign was re-launched.

The younger age of HPV-positive women aligns with global patterns showing that HPV infections are most frequent in younger women and decline with age due to viral clearance [25,26]. Lower parity among HPV-positive participants reflects previous evidence suggesting that reproductive factors influence the persistence and progression of HPV infection [27,28]. The higher proportion of single women among HPV-positive cases likely reflects behavioral risk factors related to sexual exposure [29].

HPV-16 predominated in this cohort, followed by HPV-52 and HPV-33. This distribution is consistent with international data showing HPV-16 as the leading oncogenic type worldwide and HPV-52 and HPV-33 as important contributors in Asia and Eastern Europe [30,31]. The relatively high prevalence of HPV-59 in our study also points to possible regional variation. These findings echo earlier work from Kazakhstan and Central Asia documenting the predominance of HPV-16, HPV-52, and HPV-33 in cervical disease [6,20].

HPV-16 was strongly associated with abnormal colposcopic findings, with prevalence rising stepwise across higher colposcopic grades. No other HPV genotype showed such an association after adjustment. Biopsy results confirmed HPV-16 as the only type significantly linked to CIN II/III, consistent with its well-documented oncogenic potential [32,33]. Colposcopy also proved predictive of disease severity: normal colposcopy corresponded to benign histology, while high-grade impressions were strongly associated with CIN II–III and carcinoma *in situ*, supporting previous evidence of colposcopic value in guiding biopsy.

These findings highlight the pressing need for HPV-based primary screening and genotype-specific triage in Kazakhstan, where vaccination coverage remains limited. The situation with cervical cancer screening and HPV vaccination in Kazakhstan is similar to the one in Kyrgyzstan [34]; however, some other Central Asian countries like Uzbekistan and Turkmenistan have overcome the issues and implemented effective screening and HPV vaccination years earlier with high coverage [34,35,36]. Comparing the situation with other countries from the European region of the WHO, similar issues with cervical cancer screening and HPV vaccination resulting in the high incidence of cervical cancer remain in Romania [37].

The predominance of HPV-16 among women in Kazakhstan underscores the importance of the implementation of co-testing (cervical cytology with HPV genotyping) as a national cervical cancer screening approach. While the frequent detection of HPV-52 and HPV-33 suggests that nonavalent vaccination would offer substantial additional protection. Strengthening colposcopy training and integrating HPV testing into screening pathways will be essential to reducing cervical cancer incidence in the country.

Both cytology and colposcopy in this study were within the diagnostic algorithm, but findings show their inherent performance limitations. Although being well accepted as a first-line screening tool in Kazakhstan, cytology is of moderate sensitivity for detecting cervical lesions [38,39]. Contrary to that, colposcopy is more sensitive (often 70–85%) for the diagnosis of evident lesions but has major inter-observer variability and reduced specificity (often 40–65%), as well as benign inflammatory changes being misleading [40,41].

### Strengths and Limitations

This study has several important strengths that enhance the validity and relevance of its findings. This is the first investigation in Kazakhstan to examine the correlation of HPV status, cervical cytology, colposcopic findings, and biopsy-confirmed outcomes in a non-vaccinated female population, offering timely baseline data at the launch of the national HPV vaccination program. Moreover, the relatively large sample size for HPV testing and cytology provides more robust prevalence estimates and strengthens the observed associations. The use of standardized laboratory procedures and internationally recognized diagnostic criteria for cytology, colposcopy, and histopathology ensures that the results are reliable and comparable with studies conducted in other settings.

This study also has certain limitations that should be acknowledged. First, being a single-center study may restrict the generalizability of the findings to the wider population. Second, although the overall cohort was sizable, the subset of participants who underwent biopsy was relatively small, which may have reduced the statistical power to detect associations across different HPV genotypes. Third, the absence of vaccinated participants means that vaccine effectiveness could not be evaluated, which will become an important focus as immunization coverage expands. At the current time, the proportion of HPV-vaccinated females in the country is extremely low, as during the pilot HPV vaccination campaign, only 11,000 girls were vaccinated. Fourth, despite adherence to standardized criteria, colposcopic interpretation is inherently subjective, and inter-observer variability remains a potential source of bias. The exclusion of smokers and recruitment of 396 patients instead of the planned 400 (1% is missing) should also be acknowledged as the study limitations. So, some potential biases may have an impact on the results: single-center study, age group selection (reproductive age), and exclusion of vaccinated women. The future studies should cover a larger sample size and include the cohort of vaccinated women.

## 5. Conclusions

This study examined the relationship between HPV infection status, cytology, colposcopy, and biopsy results in a cohort of unvaccinated women in Kazakhstan. We identified HPV-16 as the most common genotype, the only one consistently linked to abnormal colposcopic findings and high-grade histological lesions, emphasizing its key role in cervical cancer development. The distribution of HPV-52 and HPV-33 underscores the importance of including a broader range of genotypes in prevention efforts. These results support implementing HPV-based primary screening, genotype-specific triage, and expanding vaccination programs—especially with the nonavalent vaccine—to combat the high rate of cervical cancer in Kazakhstan. Larger multicenter studies with more biopsy data are needed to confirm these findings and evaluate vaccine effectiveness as coverage increases. Overall, the results call for urgent action to strengthen HPV screening and establish a nationwide HPV vaccination program in Kazakhstan and Central Asia.

Given the importance of the cervical cancer problem in Kazakhstan, the following recommendations based on the international best practices could be useful for the local healthcare setting:Implementation of co-testing (cervical cytology and HPV genotyping) as a cervical cancer screening;Improve cervical cancer screening coverage and participation with community education, mobile screening teams, and electronic reminders.Improve laboratory quality assurance through standardized training, external audit, and laboratory accreditation;Integrate screening and vaccination registries for patient tracking;Increase public acceptance of HPV vaccination by implementing consistent nationwide information/communication interventions.

## Figures and Tables

**Figure 1 vaccines-13-01151-f001:**
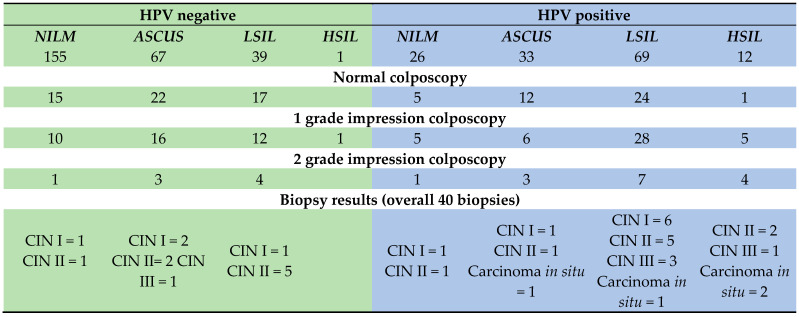
Patients’ recruitment and distribution.

**Figure 2 vaccines-13-01151-f002:**
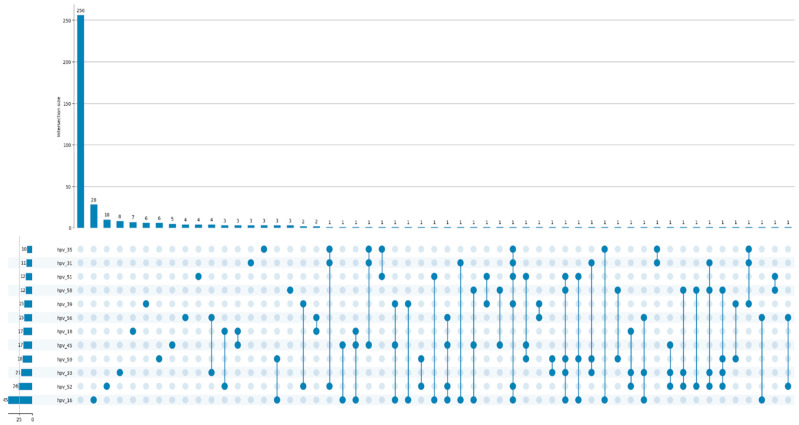
Prevalence of HPV type among the study cohort.3.3. Correlation of HPV status, cytology, colposcopy and biopsy results.

**Figure 3 vaccines-13-01151-f003:**
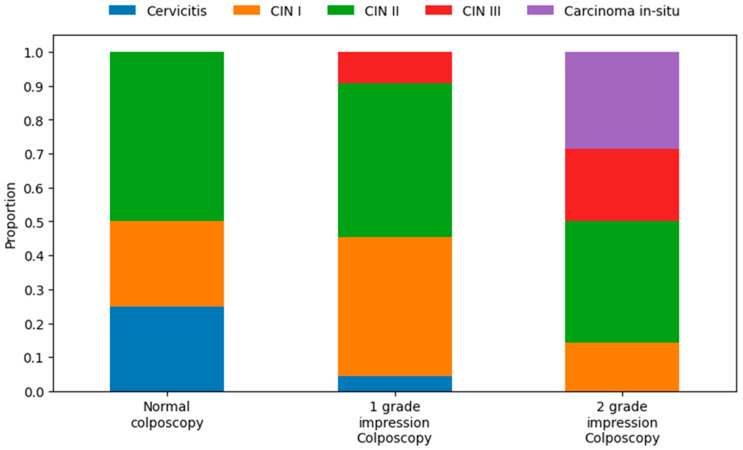
Correlation of colposcopy impression and biopsy histological outcomes.

**Table 1 vaccines-13-01151-t001:** Socio-demographic data of study subjects.

Variable	Statistics	HPV Negative	HPV Positive
Age	Mean ± SD	35.25 ± 6.18	33.17 ± 6.60
	Median [Q1–Q3]	34.96 [31.60–40.00]	33.86 [27.92–37.48]
	Min–Max	19.93–45.88	19.90–46.03
	Statistic (p)	t = 3.13	*p* = 0.0019
Ethnicity	Kazakh	236 (91.8%)	125 (89.9%)
	Russian	12 (4.7%)	8 (5.8%)
	Others	9 (3.5%)	6 (4.3%)
	Statistic (p)	Chi-square: 11.09	*p* = 0.4354
Education	Incomplete secondary	2 (0.8%)	1 (0.7%)
	Secondary professional	42 (16.3%)	17 (12.2%)
	Bachelor	178 (69.3%)	98 (70.5%)
	Masters	32 (12.5%)	21 (15.1%)
	PhD	3 (1.2%)	2 (1.4%)
	Statistic (p)	Chi-square: 1.58	*p* = 0.8130
Marital status	Single	48 (18.7%)	43 (30.9%)
	Married	184 (71.6%)	77 (55.4%)
	Divorced	22 (8.6%)	18 (12.9%)
	Widow	3 (1.2%)	1 (0.7%)
	Statistic (p)	Chi-square: 11.39	*p* = 0.0098
BMI	Mean ± SD	23.85 ± 4.62	23.36 ± 4.56
	Median [Q1–Q3]	23.14 [20.57–26.35]	22.31 [20.26–25.56]
	Min–Max	15.78–43.26	14.88–42.80
	Statistic (p)	t = 1.03	*p* = 0.3037
Number of sexual partners	Mean ± SD	1.57 ± 1.79	2.18 ± 5.42
	Median [Q1–Q3]	1.00 [1.00–1.00]	1.00 [1.00–2.00]
	Min–Max	0.00–20.00	0.00–60.00
	Statistic (p)	t = −1.65	*p* = 0.0991
Number of pregnancies	Mean ± SD	2.38 ± 1.91	1.51 ± 1.60
	Median [Q1–Q3]	2.00 [1.00–4.00]	1.00 [0.00–2.00]
	Min–Max	0.00–8.00	0.00–9.00
	Statistic (p)	U = 22,720.00	*p* = 0.0000
Number of deliveries	Mean ± SD	1.74 ± 1.39	1.20 ± 1.26
	Median [Q1–Q3]	2.00 [0.00–3.00]	1.00 [0.00–2.00]
	Min–Max	0.00–5.00	0.00–5.00
	Statistic (p)	t = 3.81	*p* = 0.0002
Number of abortions	Mean ± SD	0.36 ± 0.74	0.13 ± 0.43
	Median [Q1–Q3]	0.00 [0.00–0.00]	0.00 [0.00–0.00]
	Min–Max	0.00–4.00	0.00–3.00
	Statistic (p)	U = 20,258.00	*p* = 0.0011
Abortions	0	198 (77.0%)	125 (89.9%)
	1	59 (23.0%)	14 (10.1%)
	Statistic (p)	Chi-square: 9.12	*p* = 0.0025
Barrier contraception	0	182 (70.8%)	82 (59.0%)
	1	75 (29.2%)	57 (41.0%)
	Statistic (p)	Chi-square: 5.16	*p* = 0.0232
Hormonal contraception	0	249 (96.9%)	136 (97.8%)
	1	8 (3.1%)	3 (2.2%)
	Statistic (p)	Fisher: 0.69	*p* = 0.7536
Any contraception	0	174 (67.7%)	79 (56.8%)
	1	83 (32.3%)	60 (43.2%)
	Statistic (p)	Chi-square: 4.16	*p* = 0.0414
	STI negative	53 (20.6%)	35 (25.2%)
STI status	STI positive	204 (79.4%)	104 (74.8%)
	Statistic (p)	Chi-square: 0.84	*p* = 0.3604

**Table 2 vaccines-13-01151-t002:** Distribution of High-Risk HPV Genotypes According to Colposcopic Findings.

Variable	Statistic	Overall	Normal Colposcopy	1 Grade Impression Colposcopy	2 Grade Impression Colposcopy	Chi^2^ Statistic	Chi^2^ *p*-Value (FDR-BH)	Cramér’s V	CATT *p*-Value (FDR-BH)
HPV-16	Negative	162 (81.8%)	87 (90.6%)	65 (80.2%)	10 (47.6%)	21.65	0.000259	0.331	0.000226
	Positive	36 (18.2%)	9 (9.4%)	16 (19.8%)	11 (52.4%)				
	Group 0 vs. 1	OR = 0.42 [0.17–1.01], p (Holm) = 0.0541							
	Group 0 vs. 2	OR = 0.09 [0.03–0.28], p (Holm) = 0.0001							
	Group 1 vs. 2	OR = 0.22 [0.08–0.62], p (Holm) = 0.0095							
HPV-18	Negative	185 (93.4%)	90 (93.8%)	75 (92.6%)	20 (95.2%)	0.22	0.969078	0.033	0.974141
	Positive	13 (6.6%)	6 (6.2%)	6 (7.4%)	1 (4.8%)				
HPV-31	Negative	189 (95.5%)	92 (95.8%)	77 (95.1%)	20 (95.2%)	0.06	0.969078	0.018	0.974141
	Positive	9 (4.5%)	4 (4.2%)	4 (4.9%)	1 (4.8%)				
HPV-33	Negative	182 (91.9%)	89 (92.7%)	72 (88.9%)	21 (100.0%)	2.93	0.740104	0.122	0.974141
	Positive	16 (8.1%)	7 (7.3%)	9 (11.1%)	0 (0.0%)				
HPV-35	Negative	193 (97.5%)	93 (96.9%)	81 (100.0%)	19 (90.5%)	6.42	0.262658	0.180	0.974141
	Positive	5 (2.5%)	3 (3.1%)	0 (0.0%)	2 (9.5%)				
	Group 0 vs. 1	OR = 6.10 [0.31–119.90], p (Holm) = 0.4377							
	Group 0 vs. 2	OR = 0.31 [0.05–1.96], p (Holm) = 0.4377							
	Group 1 vs. 2	OR = 0.05 [0.00–1.04], p (Holm) = 0.1223							
HPV-39	Negative	187 (94.4%)	90 (93.8%)	77 (95.1%)	20 (95.2%)	0.17	0.969078	0.029	0.974141
	Positive	11 (5.6%)	6 (6.2%)	4 (4.9%)	1 (4.8%)				
HPV-45	Negative	187 (94.4%)	92 (95.8%)	76 (93.8%)	19 (90.5%)	1.04	0.857886	0.073	0.925942
	Positive	11 (5.6%)	4 (4.2%)	5 (6.2%)	2 (9.5%)				
HPV-51	Negative	189 (95.5%)	93 (96.9%)	77 (95.1%)	19 (90.5%)	1.67	0.803843	0.092	0.925942
	Positive	9 (4.5%)	3 (3.1%)	4 (4.9%)	2 (9.5%)				
HPV-52	Negative	181 (91.4%)	89 (92.7%)	74 (91.4%)	18 (85.7%)	1.07	0.857886	0.074	0.925942
	Positive	17 (8.6%)	7 (7.3%)	7 (8.6%)	3 (14.3%)				
HPV-56	Negative	187 (94.4%)	91 (94.8%)	76 (93.8%)	20 (95.2%)	0.11	0.969078	0.023	0.974141
	Positive	11 (5.6%)	5 (5.2%)	5 (6.2%)	1 (4.8%)				
HPV-59	Negative	183 (92.4%)	89 (92.7%)	73 (90.1%)	21 (100.0%)	2.34	0.740104	0.109	0.974141
	Positive	15 (7.6%)	7 (7.3%)	8 (9.9%)	0 (0.0%)				
HPV-58	Negative	190 (96.0%)	92 (95.8%)	79 (97.5%)	19 (90.5%)	2.15	0.740104	0.104	0.974141
	Positive	8 (4.0%)	4 (4.2%)	2 (2.5%)	2 (9.5%)				
HPV status	Negative	98 (49.5%)	53 (55.2%)	38 (46.9%)	7 (33.3%)	3.66	0.693801	0.136	0.388964
	Positive	100 (50.5%)	43 (44.8%)	43 (53.1%)	14 (66.7%)				

**Table 3 vaccines-13-01151-t003:** Distribution of High-Risk HPV Genotypes According to Biopsy results.

Variable	Status	Overall	CIN I	CIN II	CIN III	Carcinoma *In Situ*	Fisher’s Exact *p*-Value (MC)	Fisher’s Exact *p*-Value (MC) (FDR-BH)	Cramér’s V	CATT *p*-Value (FDR-BH)
HPV-16	Negative	24 (63.2%)	10 (83.3%)	12 (70.6%)	1 (20.0%)	1 (25.0%)	0.031197	0.181982	0.487	0.078658
	Positive	14 (36.8%)	2 (16.7%)	5 (29.4%)	4 (80.0%)	3 (75.0%)				
	CIN I vs. CIN II	OR = 0.48 [0.08–3.03], p (Holm) = 1.0000								
	CIN I vs. CIN III	OR = 0.05 [0.00–0.72], p (Holm) = 0.1658								
	CIN I vs. CIS	OR = 0.07 [0.00–1.02], p (Holm) = 0.3159								
	CIN II vs. CIN III	OR = 0.10 [0.01–1.18], p (Holm) = 0.4634								
	CIN II vs. CIS	OR = 0.14 [0.01–1.68], p (Holm) = 0.7584								
	CIN III vs. CIS	OR = 1.33 [0.06–31.12], p (Holm) = 1.0000								
HPV-18	Negative	36 (94.7%)	11 (91.7%)	17 (100.0%)	5 (100.0%)	3 (75.0%)	0.208579142	0.625737426	0.347	0.836420
	Positive	2 (5.3%)	1 (8.3%)	0 (0.0%)	0 (0.0%)	1 (25.0%)				
HPV-33	Negative	35 (92.1%)	10 (83.3%)	16 (94.1%)	5 (100.0%)	4 (100.0%)	0.837216278	1	0.237	0.847951
	Positive	3 (7.9%)	2 (16.7%)	1 (5.9%)	0 (0.0%)	0 (0.0%)				
HPV-35	Negative	37 (97.4%)	12 (100.0%)	16 (94.1%)	5 (100.0%)	4 (100.0%)	1	1	0.183	0.847951
	Positive	1 (2.6%)	0 (0.0%)	1 (5.9%)	0 (0.0%)	0 (0.0%)				
HPV-39	Negative	36 (94.7%)	11 (91.7%)	16 (94.1%)	5 (100.0%)	4 (100.0%)	0.342365763	0.732526747	0.140	0.847951
	Positive	2 (5.3%)	1 (8.3%)	1 (5.9%)	0 (0.0%)	0 (0.0%)				
HPV-45	Negative	35 (92.1%)	12 (100.0%)	16 (94.1%)	5 (100.0%)	2 (50.0%)	0.04549545	0.181981802	0.545	0.118364
	Positive	3 (7.9%)	0 (0.0%)	1 (5.9%)	0 (0.0%)	2 (50.0%)				
	CIN I vs. CIN II	OR = 0.44 [0.02–11.74], p (Holm) = 1.0000								
	CIN I vs. CIS	OR = 0.04 [0.00–1.11], p (Holm) = 0.2500								
	CIN II vs. CIN III	OR = 1.00 [0.04–28.30], p (Holm) = 1.0000								
	CIN II vs. CIS	OR = 0.06 [0.00–1.04], p (Holm) = 0.3188								
	CIN III vs. CIS	OR = 0.09 [0.00–2.68], p (Holm) = 0.5000								
HPV-51	Negative	35 (92.1%)	11 (91.7%)	16 (94.1%)	5 (100.0%)	3 (75.0%)	0.620437956	0.930656934	0.237	0.836420
	Positive	3 (7.9%)	1 (8.3%)	1 (5.9%)	0 (0.0%)	1 (25.0%)				
HPV-52	Negative	36 (94.7%)	11 (91.7%)	16 (94.1%)	5 (100.0%)	4 (100.0%)	1	1	0.140	0.847951
	Positive	2 (5.3%)	1 (8.3%)	1 (5.9%)	0 (0.0%)	0 (0.0%)				
HPV-56	Negative	36 (94.7%)	10 (83.3%)	17 (100.0%)	5 (100.0%)	4 (100.0%)	0.366263374	0.732526747	0.347	0.847951
	Positive	2 (5.3%)	2 (16.7%)	0 (0.0%)	0 (0.0%)	0 (0.0%)				
HPV-59	Negative	36 (94.7%)	11 (91.7%)	16 (94.1%)	5 (100.0%)	4 (100.0%)	1	1	0.140	0.847951
	Positive	2 (5.3%)	1 (8.3%)	1 (5.9%)	0 (0.0%)	0 (0.0%)				
HPV-58	Negative	36 (94.7%)	12 (100.0%)	17 (100.0%)	5 (100.0%)	2 (50.0%)	0.00879912	0.105589441	0.687	0.118364
	Positive	2 (5.3%)	0 (0.0%)	0 (0.0%)	0 (0.0%)	2 (50.0%)				
	CIN I vs. CIS	OR = 0.04 [0.00–1.11], p (Holm) = 0.1000								
	CIN II vs. CIS	OR = 0.03 [0.00–0.78], p (Holm) = 0.0857								
	CIN III vs. CIS	OR = 0.09 [0.00–2.68], p (Holm) = 0.1667								
HPV status	Negative	13 (34.2%)	4 (33.3%)	8 (47.1%)	1 (20.0%)	0 (0.0%)	0.462653735	0.793120688	0.315	0.836420
	Positive	25 (65.8%)	8 (66.7%)	9 (52.9%)	4 (80.0%)	4 (100.0%)				

## Data Availability

The data can be shared up on reasonable request to the project PI, Professor Talshyn Ukybassova (talshynu@yandex.ru).

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
