# Peer review of "Correlation of HPV Status with Colposcopy and Cervical Biopsy Results Among Non-Vaccinated Women: Findings from a Tertiary Care Hospital in Kazakhstan"

_vaccines, 2025, doi:10.3390/vaccines13111151_

Round 1
Reviewer 1 Report
Comments and Suggestions for Authors
Talshyn Ukybassova et al. report in their manuscript the correlation they found between colposcopy and cervical biopsy results and human papilloma virus infection status of unvaccinated women in Kazakhstan.
Human papilloma virus (HPV) infections have been identified as being often at the origin of cervical cancer in women. In Kazakhstan the vaccination rate agains HPV is still very low and the prevalence of cervical cancer is high.
396 women participated in the study. 140 had a positive HPV status. 198 women were investigated by colposcopy because of abnormal cervical cytology results. From 40 women biopsies were taken. The histological grading of those 40 revealed that 12 had CIN I, 12 had CIN II, 10 had CIN III and 4 had carcinoma in situ.
From those that were HPV infected most of them were of HPV-16 type. The follow-ups were HPV-52 and HPV-33.
When analysing cases where biopsies were taken and an HPV infection was positive, HPV-16 had a higher than random association with CIN II and CIN III types. HPV-45 and HPV-58 were seen more often than expected in the higher CIN histological grades. But since both HPV types are quite rare this association was not statistically significant.
I think this manuscript represents a well done study between the association between specific HPV gene-types and cervial lesions in women that can be precursors for cervical cancer.
The abstract of the article summarises the findings in an understandable form.
The introduction describes the medical context in Kazakhstan well.
The study group is reasonably large for a single-hospital study. The participants were well selected and significantly interrogated about their background. The investigations are well described and the findings classified based on World Health Organisation standards. The results are transparently reported in the various tables and well described within the text.
I think the conclusions are supported by the data the authors present.
There is nothing I can propose to improve the article.
In my eyes the manuscript can be published as it is.
Author Response
Dear Reviewer,
thank you for the detailed review of our manuscript and appreciation of our work.
Reviewer 2 Report
Comments and Suggestions for Authors
- Lines 41-43: There is a more recent publication on GLOBOCAN data. Also, consider citing some figures (global age-standardized rates).
- Lines 43-46: Provide data on mortality too, it makes sense to do so to strengthen the case for the need for early detection efforts (and consequently treatment). Also - very briefly mention the trends over time for the reader to get the full grasp of the situation with cervical cancer burden in Kazakhstan.
- Lines 47-54: Indicate in which year the screening was introduced. Provide specific details on coverage - including rates and any discrepancies between urban and rural areas (if available); state whether it is free.
- Lines 57-58: Is it mandatory, is it free?
- Lines 61-62: State the types.
- Lines 62-63: Even sooner for HPV incidence, and can take a bit longer for cervical cancer incidence (but early signs can be seen after a decade).
- Lines 69-71: And has any previous study attempted to assess what your study has? This seems to have been left slightly vague here.
- Methods: It is not clear - given the differences in age for inclusion criteria in this study and age targeted for screening - did the study population here include women who were invited for screening, or did these women have any symptoms and that is why they saw their doctor?
- Exclusion criteria - what if they had extensive surgeries on genital organs in the past more than 45 days ago, or conization, or previously had a cervical lesion removed or had cervical cancer?
- Why were women who were smokers excluded? This could have been appropriately addressed in the analyses, and there seems to be no methodological justification to exclude smokers. This could significantly affect the results, reduce generalizability, underestimate the estimates, even if the prevalence of smoking is relatively low in women - this introduces selection bias and makes the findings biased.
- Methods: There is no 'cross-sectional cohort study' - so state the design correctly. You may have done a cross-sectional study in a cohort of women, but how is this a cohort study?
- Methods: Provide details on study setting, study population, procedures - how were women recruited, how were they found and checked against incluson/exclusion criteria, how many women were seen in that period and how many of that number were included - provide a response rate and participation rate, how many were excluded (with reasons for exclusion), how many refused to participate, how mny had missing data and how was missing data handled.
- Methods: Provide sample size calculation.
- Methods: Describe how the data were collected, and what information was gathered from participants, and define all variables.
- Line 162: 396 or 402 how the Abstract states?
- Line 163: Was not the age of 45 and over an exclusion criteria?
- Line 235: Again, check the study design.
- Lines 237-239: Is it fully appropriate to consider this in such a context given that you have stated that the HPV vaccination has been introduced for barely a year? Everything needs to be taken into matter.
- Lines 261-262: Why not provide and overview of real-word experience around the world and in similar countries with different approaches to screening?
- Discussion is way too brief, is not comprehensive at all, lacks depth in comparisons and explanations, as well as indications for future research and implications for practice.
- Limitations - some of these could have been mitigated by properly planning the study and its sampl, based on the e.g. rate of biopsies. Also, excluding smokers lacks any justification and interpretation of its possible implications on findings and their applicability in a broader setting.
Author Response
Dear Reviewer,
Thank you very much for the detailed review of our manuscript. We appreciate your time, efforts,
valuable comments and suggestions that helped us to improve the text quality. Please find below our
point-by-point responses for all your comments.
Comments and Suggestions for Authors
- Lines 41-43: There is a more recent publication on GLOBOCAN data. Also, consider citing some figures (global age-standardized rates).
Response – Thank you for the comment. The most recent GLOBACAN and GBD data are cited.
- Lines 43-46: Provide data on mortality too, it makes sense to do so to strengthen the case for the need for early detection efforts (and consequently treatment). Also - very briefly mention the trends over time for the reader to get the full grasp of the situation with cervical cancer burden in Kazakhstan.
Response - Thank you for the comment. The mortality is provided according to GLOBACAN data, and mentioned the trends in Kazakhstan.
- Lines 47-54: Indicate in which year the screening was introduced. Provide specific details on coverage - including rates and any discrepancies between urban and rural areas (if available); state whether it is free.
Response - Thank you for the comment. Specifics of the national cervical cancer screening are provided. Please see the revised text.
- Lines 57-58: Is it mandatory, is it free?
Response - Thank you for the comment. The screening is included in the national vaccination calendar and it is free of charge.
- Lines 61-62: State the types.
Response - Thank you for the comment. The types are stated.
- Lines 62-63: Even sooner for HPV incidence, and can take a bit longer for cervical cancer incidence (but early signs can be seen after a decade).
Response - Thank you for the comment.
- Lines 69-71: And has any previous study attempted to assess what your study has? This seems to have been left slightly vague here.
Response - Thank you for the comment. There were no previous studies in Kazakhstan assessing the correlation between HPV status, cervical cytology results, colposcopy and biopsy results among our female population.
- Methods: It is not clear - given the differences in age for inclusion criteria in this study and age targeted for screening - did the study population here include women who were invited for screening, or did these women have any symptoms and that is why they saw their doctor?
Response – Thank you for the comment. All women were recruited among those attending a routine screening.
- Exclusion criteria - what if they had extensive surgeries on genital organs in the past more than 45 days ago, or conization, or previously had a cervical lesion removed or had cervical cancer?
Response - Thank you for the comment. Only women with an intact cervix without major gynecological surgeries were included.
Why were women who were smokers excluded? This could have been appropriately addressed in the analyses, and there seems to be no methodological justification to exclude smokers. This could significantly affect the results, reduce generalizability, underestimate the estimates, even if the prevalence of smoking is relatively low in women - this introduces selection bias and makes the findings biased.
Response – Thank you for the comment. Smoking serves as one of the risk factors for cancer in general and cervical cancer in particular, therefore smokers were not included. Actually, the percentage of smokers among the women population is very small in Kazakhstan and that won’t affect the results.
- Methods: There is no 'cross-sectional cohort study' - so state the design correctly. You may have done a cross-sectional study in a cohort of women, but how is this a cohort study?
Response - Thank you for the comment. Corrected. Please see the revised manuscript.
- Methods: Provide details on study setting, study population, procedures - how were women recruited, how were they found and checked against incluson/exclusion criteria, how many women were seen in that period and how many of that number were included - provide a response rate and participation rate, how many were excluded (with reasons for exclusion), how many refused to participate, how mny had missing data and how was missing data handled.
Response - Thank you for the comment. The requested details were included in the methods section.
- Methods: Provide sample size calculation.
Response – Thank you for the comment. The sample size calculation was included in the methods text.
- Methods: Describe how the data were collected, and what information was gathered from participants, and define all variables.
Response - Thank you for the comment. The details of the data collected and variables are included in the methods section. Please see the revised manuscript.
- Line 162: 396 or 402 how the Abstract states?
Response - Thank you for the comment. The typo was resolved.
- Line 163: Was not the age of 45 and over an exclusion criteria?
Response - Response - Thank you for the comment. The typo was resolved.
- Line 235: Again, check the study design.
Response - Response - Thank you for the comment. The typo was resolved.
- Lines 237-239: Is it fully appropriate to consider this in such a context given that you have stated that the HPV vaccination has been introduced for barely a year? Everything needs to be taken into matter.
Response - Response - Thank you for the comment. Please see the revised text.
- Discussion is way too brief, is not comprehensive at all, lacks depth in comparisons and explanations, as well as indications for future research and implications for practice.
Response - Response - Thank you for the comment. The discussion was expanded. Please see the revised version of the manuscript.
- Limitations - some of these could have been mitigated by properly planning the study and its sampl, based on the e.g. rate of biopsies. Also, excluding smokers lacks any justification and interpretation of its possible implications on findings and their applicability in a broader setting.
Response - Response - Thank you for the comment. These issues will be addressed in our future studies.

Reviewer 3 Report
Comments and Suggestions for Authors
The manuscript addresses a relevant public health issue (cervical cancer, HPV, vaccination). The results are clear and supported by numerical data. However, to enhance the quality of this paper, it would be beneficial to consider these issues:
- Title: needs revision to be clear and concise without acronyms; the region should be mentioned, “Astana”.
Abstract:
Conclusion too general
"Nonavalent vaccination" could be more precise: "nonavalent" (9-valent) or "multivalent" should be clearly stated.
Keywords need revision, including the country.
Introduction
- Authors consider providing insight into the global epidemiological burden: incidence, mortality, and role in female cancers.
- Emphasize that cervical cancer is almost entirely preventable through screening and vaccination.
- Please rewrite the last paragraph by indicating the research question and stating clearly the objectives of this study.
Methodology
The term "cross-sectional cohort study" is paradoxical. A cohort is prospective, while a cross-sectional study is a snapshot.
It's important to clarify: cross-sectional study? prospective cohort? observational study?
A subsection about population is required
The recruitment is not indicated
Inclusion and exclusion criteria need revision. why only non-vaccinated women?
Results
Table 1 could be improved; the statistics must be indicated in an additional column.
Please avoid using acronyms in subheadings
Discussion
- Revise the Discussion section so that your first paragraph summarizes the objective of your case study.
- Although the associations between HPV16, colposcopy, and histology are well developed, the correlation between HPV status and cytology results, although mentioned in the study objective, is not discussed. Similarly, the respective performance of cytology and colposcopy is not compared, although this would strengthen the justification for primary HPV-based screening. Add a critical analysis of the sensitivity/specificity of cytology vs. colposcopy, and justify the superiority of HPV screening.
- The discussion clearly highlights the need for HPV screening and vaccination, but it does not compare the results to WHO recommendations or guidelines (e.g., ASCCP). The absence of this comparison prevents the results from being positioned in a health policy perspective. Add a section comparing Kazakhstan's current strategy to international recommendations, and suggest adaptations (HPV primary screening, genotypic triage, nonavalent vaccination).
- The discussion concludes on the importance of vaccination and screening, but does not propose concrete implementation strategies (staff training, accessibility, awareness, pilot programs). Nor does it consider complementary avenues of research (longitudinal studies, monitoring of viral clearance, study of genotyping performance).
- Further discussion of potential biases (vaccinated vs non-vaccinated, unmeasured confounders) and clinical implications would enhance the value of the article.
- The limitations should include potential biases, and future research directions are necessary to strengthen the discussion.
- Conclusion
The conclusion could be more impactful if it included concrete clinical recommendations based on the study's findings. It would also be relevant to add suggestions for future research.
Author Response
Dear Reviewer,
Thank you very much for the detailed review of our manuscript. We appreciate your time, efforts,
valuable comments and suggestions that helped us to improve the text quality. Please find below our
point-by-point responses for all your comments.
Comments and Suggestions for Authors
The manuscript addresses a relevant public health issue (cervical cancer, HPV, vaccination). The results are clear and supported by numerical data. However, to enhance the quality of this paper, it would be beneficial to consider these issues:
1.- Title: needs revision to be clear and concise without acronyms; the region should be mentioned, “Astana”.
Response – Thank you for the comment. Since the UMC clinics are accumulating patients from various regions of the country, as explained in the revised methods section, we would like to keep “Kazakhstan” in the title because patients recruited to the study came from different regions of the country, not limited to Astana. Thank you for your understanding.
2.Abstract:
Conclusion too general
"Nonavalent vaccination" could be more precise: "nonavalent" (9-valent) or "multivalent" should be clearly stated.
Response – Thank you for the comment. The abstract was revised. However, we would like to avoid mentioning the vaccine brand names to avoid the appearance of lobbying or conflict of interest. Therefore, we prefer to keep a “nonavalent” vaccine.
3.Keywords need revision, including the country.
Response – Thank you for the comment. Revised.
Introduction
4.- Authors consider providing insight into the global epidemiological burden: incidence, mortality, and role in female cancers.
Response – Thank you for the comment. A more updated epidemiology is provided in the updated introduction. Please see the revised text.
5.- Emphasize that cervical cancer is almost entirely preventable through screening and vaccination.
Response – Thank you for the comment. Included a statement. Please see the revised text.
6.- Please rewrite the last paragraph by indicating the research question and stating clearly the objectives of this study.
Response - Thank you for the comment. The paragraph is revised Please see the revised text.
Methodology
7.The term "cross-sectional cohort study" is paradoxical. A cohort is prospective, while a cross-sectional study is a snapshot.
It's important to clarify: cross-sectional study? prospective cohort? observational study?
Response – thank you for the comment. The issue was revised.
8.A subsection about population is required
Response – thank you for the comment. Included. Please see the revised methods section.
9.The recruitment is not indicated
Response – thank you for the comment. Included. Please see the revised methods section.
10.Inclusion and exclusion criteria need revision. why only non-vaccinated women?
Response – thank you for the comment. Details are included. Please see the revised methods section.
We included only non-vaccinated females, as generally, HPV vaccination was not available in Kazakhstan before 2024. Only a minor proportion of girls got the vaccine during the pilot HPV vaccination program in 2013. However, we plan to perform a study including the vaccinated population a few years later, when the national HPV vaccination program, which was re-launched in 2024, will cover a sufficient proportion of female in Kazakhstan.
Results
11.Table 1 could be improved; the statistics must be indicated in an additional column.
Response – Thank you for the comment. However, we are not sure what was requested. The statistics is indicated in the separate column (column 2) as requested by the reviewer.
12.Please avoid using acronyms in subheadings
Response – Thank you for the comment. Acronyms were removed from subheadings. Full spelling is provided.
Discussion
13.- Revise the Discussion section so that your first paragraph summarizes the objective of your case study.
Response – Thank you for the comment. The discussion part was revised. See the re-submitted manuscript.
14- Although the associations between HPV16, colposcopy, and histology are well developed, the correlation between HPV status and cytology results, although mentioned in the study objective, is not discussed. Similarly, the respective performance of cytology and colposcopy is not compared, although this would strengthen the justification for primary HPV-based screening. Add a critical analysis of the sensitivity/specificity of cytology vs. colposcopy, and justify the superiority of HPV screening.
Response – Thank you for the comment. The values of the methods used are discussed.
15.- The discussion clearly highlights the need for HPV screening and vaccination, but it does not compare the results to WHO recommendations or guidelines (e.g., ASCCP). The absence of this comparison prevents the results from being positioned in a health policy perspective. Add a section comparing Kazakhstan's current strategy to international recommendations, and suggest adaptations (HPV primary screening, genotypic triage, nonavalent vaccination).
Response – Thank you for the comment. However, the aim of this study was to investigate correlations between cervical cytology, HPV status, and colposcopy results. This study did not aim to compare the national and international guidelines on cervical cancer prevention. However, at the end of the manuscript, we included the recommendations for the national screening based on the international guidelines.
16.- The discussion concludes on the importance of vaccination and screening, but does not propose concrete implementation strategies (staff training, accessibility, awareness, pilot programs). Nor does it consider complementary avenues of research (longitudinal studies, monitoring of viral clearance, study of genotyping performance).
Response – Thank you for the comment. However, the aim of this study was to investigate correlations between cervical cytology, HPV status, and colposcopy results. This study did not aim to develop a strategy for vaccination. Thank you for the recommendation. Concrete implementation strategies (staff training, accessibility, awareness, pilot programs) for HPV vaccination in Kazakhstan will be provided in our next manuscript, which is under development now.
17.- Further discussion of potential biases (vaccinated vs non-vaccinated, unmeasured confounders) and clinical implications would enhance the value of the article.
Response – Thank you for the comment. It is reported in the strengths and limitations section.
18.- The limitations should include potential biases, and future research directions are necessary to strengthen the discussion.
Response – Thank you for the comment. The strengths and limitations section was expanded.
- Conclusion
19.The conclusion could be more impactful if it included concrete clinical recommendations based on the study's findings. It would also be relevant to add suggestions for future research.
Response – Thank you for the comment. The conclusion was revised.

Reviewer 4 Report
Comments and Suggestions for Authors
This study evaluated the correlation between HPV infection and colposcopic findings. The topic is relevant however the methodology of this study is not clear. It is not clear whether this is a prospective or a retrospective study. In addition, the authors do not explain why the duration of this study was only 4 months. The patient inclusion should be presented in a flowchart. Grade 1 and grade 2 colposcopy impressions are not defined in the methods section. Perhaps a scoring system (Swede score, Reid index...) would be more appropriate.
Author Response
Dear Reviewer,
Thank you very much for the detailed review of our manuscript. We appreciate your time, efforts,
valuable comments and suggestions that helped us to improve the text quality. Please find below our
point-by-point responses for all your comments.
Comments and Suggestions for Authors
This study evaluated the correlation between HPV infection and colposcopic findings. The topic is relevant however the methodology of this study is not clear. It is not clear whether this is a prospective or a retrospective study. In addition, the authors do not explain why the duration of this study was only 4 months. The patient inclusion should be presented in a flowchart. Grade 1 and grade 2 colposcopy impressions are not defined in the methods section. Perhaps a scoring system (Swede score, Reid index...) would be more appropriate.
Response – thank you for the comment. The methods section was revised, and the details of the recruitment, sample size, and variables were included. Please see the revised manuscript with track changes highlighting the amendments made to the text.
Round 2
Reviewer 2 Report
Comments and Suggestions for Authors
The Authors have provided responses, however not all questions were answered and provided answers were not satisfactory. Also, I would especially underline that the responses need to be appropriately entered in the text and not only in the response letter, and all of the information needs to be backed up with citations of corresponding appropriate references.
- Comment asking whether the vaccine is free was responded to only in the cover letter and no clarification was included in the paper.
- Comment regarding the sentence "The effect of vaccination in terms of decreasing the incidence of HPV infection and cervical cancer will be detectable in the coming 10 years" - was meant for authors to check and correct their statement as appropriate - this comment was left unaddressed, this was not my remark for the author but for the paper, which is the case for all of my comments. A reference needs to be added in the text.
- The response for study population is unclear. The Authors stated in their response that the women who were included in the study were women attending "routine cervical screening". An explanation stating if women are invited attend routine cervical cancer screening at the tertiary care level clinic is necessary - with references backing up this.
- Continuing with the above comment - why and how did this study include women aged 18-45 years old if the Authors themselves have stated "A nationwide cytology-based cervical cancer screening program was introduced in Kazakhstan in 2017 and fully covered by the state budget. The target group is women aged 30–70 years, with the screening every four years [8]." - how is this plausible and possible? Provide explanations, clarify and align the information provided throughout the paper. Did these women attend as you have stated routine cervical cancer screening? If not - what were the reasons for them attending screening? Note - the latter is completely different from attending routine screening.
- The Authors have still not described how the study was conducted - when women came to the clinic, what was the reason for their visit if not routine screening, who asked them whether they want to participate in the study, who checked and how they verified whether they had any inclusion or exclusion criteria. If out of 753 women 396 agreed and the rest refused - that means that all 753 who were invited to participate fulfilled the inclusion criteria - but how many were there who were "screened" and did not fulfill these criteria during the study period? No answer was provided on response and participation rate.
- Sample size calculation was added but did not state the specific parameters that the Authors included in their calculation - the coefficient, proportion, error, which need to be stated explicitly.
- The Authors stated that the age of included participants ranging to 46, despite including persons up to the age of 45 was a typo and they changed it in the text. But how can it be a typo if the data in the Table 1 clearly shows this?
- The first paragraph of the Discussion was revised but this paragraph is usually focused on the results of the study itself, rather than what was previously shown - this information is given further on.
- Limitations of the study still do not mention excluding smokers. Also, there is no mention that the required minimum sample size as you have provided was not reached - and its possible implications.
- The response to my comment about smoking being a criterion for exclusion and my comment for surgeries in 45 days is debatable. It is well-known that smoking is an established risk factor for cervical cancer. Nowhere in the entire work is hysterectomy, whether total or partial, mentioned as a factor for determining the eligible population. Because of all this, the aforementioned circumstances of selecting the study population can lead to potential sources of bias, primarily selection bias. The authors did not adequately address my comments regarding the selection of participants in this study in their responses.
Author Response
Dear Reviewer, Thank you very much for the detailed review of our manuscript. We appreciate a lot your time, efforts, valuable comments, and suggestions that helped us to improve the quality of the text. Please find below our response to your comments.

Reviewer 3 Report
Comments and Suggestions for Authors
All issues have been adressed by authors. No further comments.
Author Response
Thank you!